# Ghrelin as a Biomarker of “Immunometabolic Depression” and Its Connection with Dysbiosis

**DOI:** 10.3390/nu15183960

**Published:** 2023-09-13

**Authors:** Agata Gajewska, Dominik Strzelecki, Oliwia Gawlik-Kotelnicka

**Affiliations:** 1Faculty of Medicine, Medical University of Lodz, 92-216 Lodz, Poland; agata.gajewska@stud.umed.lodz.pl; 2Department of Affective and Psychotic Disorders, Medical University of Lodz, 92-216 Lodz, Poland; dominik.strzelecki@umed.lodz.pl

**Keywords:** ghrelin, depression, metabolic syndrome, obesity, microbiota

## Abstract

Ghrelin, a gastrointestinal peptide, is an endogenous ligand of growth hormone secretagogue receptor 1a (GHSR1a), which is mainly produced by X/A-like cells in the intestinal mucosa. Beyond its initial description as a growth hormone (GH) secretagogue stimulator of appetite, ghrelin has been revealed to have a wide range of physiological effects, for example, the modulation of inflammation; the improvement of cardiac performance; the modulation of stress, anxiety, taste sensation, and reward-seeking behavior; and the regulation of glucose metabolism and thermogenesis. Ghrelin secretion is altered in depressive disorders and metabolic syndrome, which frequently co-occur, but it is still unknown how these modifications relate to the physiopathology of these disorders. This review highlights the increasing amount of research establishing the close relationship between ghrelin, nutrition, microbiota, and disorders such as depression and metabolic syndrome, and it evaluates the ghrelinergic system as a potential target for the development of effective pharmacotherapies.

## 1. Ghrelin Characteristics

### 1.1. Origin of Ghrelin

The gastrointestinal peptide hormone ghrelin was discovered in 1999 as the endogenous ligand for the growth hormone secretagogue receptor (GHSR)1a, which can stimulate the release of growth hormone (GH) from the anterior pituitary gland [1]. Ghrelin is an acyl-peptide formed by 28 amino acids. It is primarily produced by X/A-like cells in the intestinal mucosa, but it can also be found in the pituitary gland, brain cortex, hypothalamus, adrenal gland, hippocampus, intestine, pancreas, and many other human tissues [2].

The human preproghrelin GHRL gene is localized on chromosome 3p25-26. In addition to ghrelin, it also encodes a small signal peptide and the 23-amino-acid peptide obestatin. The GHRL gene generates the 117-amino-acid preproghrelin peptide, which is then converted by enzymes into the native 28-amino-acid peptide des-acyl ghrelin (DAG) [3]. Afterwards, DAG undergoes enzymatic modification in the endoplasmic reticulum to generate acyl-ghrelin. The enzyme that catalyzes the octanoylation of ghrelin is ghrelin O-acyltransferase (GOAT), a member of the family membrane-bound O-acyl transferase (MBOAT). Octanoylation is achieved by attaching an eight-carbon medium-chain fatty acid (MCFA) (octanoate) to serine 3 of ghrelin [4,5] (Figure 1). Ghrelin is the only peptide hormone in mammals that is acylated, which is crucial for its binding to GHS-R1a [6]. The metabolic effects of ghrelin appear to be influenced by the length of the fatty acid used for acylation. GHSR1a is differentially activated by changes in fatty acid length in vitro, and these changes also affect how ghrelin impacts food intake and adiposity in vivo [7].

About 90% of circulating ghrelin is in the DAG form, with only less than 10% being acylated [8]. AG is mostly bound to larger molecules, like high-density lipoprotein, very-low-density lipoprotein, and low-density lipoprotein. DAG is more often present as a free peptide, but it can also bind to HDL [9].

DAG cannot bind with the GHS-R1a receptor, but recent evidence indicates that it behaves like a separate hormone, and it appears to have effects that are independent of AG [10] and may also affect metabolism via an unknown receptor [11,12]. DAG, for instance, promotes C2C12 skeletal muscle cell development [13], prevents muscle atrophy [14,15], provides cardioprotective effects [16], and regulates glucose metabolism [17].

The term “ghrelin” used further in this article refers to acyl-ghrelin.

### 1.2. Functions of Ghrelin

Beyond its initial association with growth hormone release and appetite regulation, acyl-ghrelin modulates many physiological pathways. The various functions of acyl-ghrelin are summarized in the table below (Table 1).

### 1.3. Regulation of Circulating Ghrelin

Numerous factors, including nutrition, hormones, and neural regulation, affect the regulation of ghrelin, but the precise mechanisms are still mostly unclear. Plasma ghrelin levels decrease to 10–30% of preoperative levels after total gastrectomy and to 50–70% after distal gastrectomy, which indicates that the X/A-like cells in the stomach are the primary source of circulating ghrelin [40] (Figure 2).

Due to the fact that ghrelin rises prior to meals and declines after, fasting enhances plasma ghrelin levels, while eating decreases them [18]. The nutritional composition of the diet also influences circulating ghrelin levels. Some studies investigated whether meals with varying macronutrient compositions but equivalent energy quantities affected postprandial ghrelin production differentially. According to the researchers, a high-protein meal suppresses hunger and increases the time between meals more than a high-fat or high-carbohydrate meal, possibly through increasing postprandial ghrelin suppression [41,42]. The proposed mediators that may be involved in the regulation of postprandial ghrelin secretion include glucose [43], insulin [44], cholecystokinin [45], GLP-1 [46], and somatostatin [47], the majority of which are gastrointestinal hormones that delay gastric emptying and have insulinotropic and anorexigenic properties [31,48].

## 2. Immunometabolic Depression

In a meta-analysis, Vancampfort et al. found that the risk of acquiring metabolic syndrome (MetS), which is diagnosed in 30% of depressed people, was 1.5 times higher in people with major depressive disorder (MDD) [49]. Depressed individuals also have an elevated risk of obesity [50,51] and insulin resistance [52,53]. The connection is still poorly understood, most likely because both metabolic syndrome and MDD are complicated, multifaceted diseases, but recent research has identified both as inflammatory disorders that involve systemic and central immune cells [54].

Researchers speculate that inflammation underlies the frequent comorbidity between metabolic syndrome and MDD, as both conditions share several inflammatory disturbances. The evidence that inflammation plays a key role in the immunopathogenesis of those illnesses is supported by the fact that levels of proinflammatory cytokines like IL-6, IL-1, and TNF-alfa are raised both in patients with MetS [55,56] and MDD [57,58] compared to healthy controls. It has been demonstrated that persistent stress, which correlates with depression and anxiety [59], raises the level of cytokines [60]. Indeed, IL-6 [61] and CRP [62] levels are associated with depression severity, predict cognitive impairments in depression [63], and are further aggravated by acute stress [64]. Additional research discovered that obese individuals, but not non-obese individuals, had favorable relationships between CRP levels and depressive mood (Depressive symptoms in obesity: Relative contribution of low-grade inflammation and metabolic health), indicating that inflammation may mediate certain types of depressive disorders that occur in conjunction with obesity and metabolic syndrome.

TNF-alfa and IL-6 levels are also significantly higher in obese individuals [65,66], and they correlate with body weight, BMI, waist and hip circumferences, and waist–hip ratio [67]. It has been shown that obesity-induced low-grade inflammation (raised IL-1, IL-6, and TNF-alfa levels) activates stress kinases, including IKK, JNK, and p38 MAPK, in muscle and fat cells. Therefore, insulin resistance occurs [68,69], which has been proven to double the risk of major depressive disorder [70]. It has been shown that weight loss reduces this increased cytokine expression in adipose tissue, reducing systemic inflammation [71]. Interestingly, patients who had previously been obese showed a reduction in inflammation after weight loss in a surgery-induced weight-loss intervention study, which correlated with significant positive changes in emotional status [72]. 

In addition to the peripheral inflammation associated with obesity, new research has focused on inflammation in the central nervous system (CNS), focusing on the hypothalamus due to its function in hunger and satiety regulation. Diet-induced obesity is widely known to be linked to hypothalamic inflammation. Many studies have been conducted to investigate the processes underlying hypothalamus inflammation, such as the Toll-like receptor 4 (TLR-4) pathway (Molecular mechanisms underlying obesity-induced hypothalamic inflammation and insulin resistance: Pivotal role of resistin/tlr4 pathways). TLR-4 activation further stimulates the IKB kinase/nuclear factor kB (NF-kB) pathway, causing the production of proinflammatory cytokines, such as IL-6, IL-8, TNF, and IL1 (Hypothalamic Inflammation and Obesity: A Mechanistic Review). Furthermore, this process enhances the inflammatory response in microglia cells, which are hypothalamic resident macrophage cells. TLR-4 activation stimulates JNK, which, in turn, promotes the production of cytokines, including interleukin (IL)-1, interleukin (IL)-6, tumor necrosis factor (TNF)-a, chemokines, and other proinflammatory compounds (Hypothalamic Inflammation and Gliosis in Obesity).

According to research, hypothalamic–pituitary–adrenal axis (HPA) activity is also dysregulated both in patients with metabolic syndrome and depression. Obesity is linked to inflammation, as seen by higher peripheral proinflammatory cytokines, which can stimulate HPA axis activity. Certain cytokines, such as IL-1β, can cross the blood–brain barrier (Passage of Cytokines across the Blood-Brain Barrier) and trigger CRH secretion in the hypothalamus, directly activating the HPA axis, resulting in remarkably increased cortisol levels (Recombinant Interleukin-6 Activates the Hypothalamic-Pituitary-Adrenal Axis in Humans).

Adipose tissue also contains increased cortisol concentrations in obese people due to impaired HPA axis function (The Hypothalamic-Pituitary-Adrenal Axis, Obesity, and Chronic Stress Exposure: Sleep and the HPA Axis in Obesity). Moreover, a recent meta-analysis found that hair cortisol levels are positively linked with anthropometric parameters, such as BMI and waist-to-hip ratio (Stress-Related and Basic Determinants of Hair Cortisol in Humans: A Meta-Analysis), which are components of metabolic syndrome. Cortisol is well known as a stress hormone, and research shows that depressive disorder is positively correlated with high cortisol levels (Stress and serum cortisol levels in major depressive disorder: a cross-sectional study).

Insulin resistance, which is linked to cognitive and mental disorders, is another metabolic abnormality that can be induced by the dysregulation of the HPA axis. Chronic increases in glucocorticoid levels block insulin’s ability to promote glucose uptake by cells, progressively leading to insulin resistance, one of the important symptoms of MS. Insulin resistance promotes the accumulation of free fatty acids in the body, which leads to obesity and diabetes (Molecular mechanisms of glucocorticoid-induced insulin resistance). According to research, acute periods of depression are associated with increased insulin resistance (Insulin resistance in depression: A large meta-analysis of metabolic parameters and variation). In individuals with depression, a laboratory measurement of insulin resistance may be useful for diagnosing the immunometabolic subtype and selecting the best treatment.

## 3. Ghrelin and Its Link to Immunometabolic Depression

The complex link between stress, mood, and food consumption is the subject of growing research, and several studies point to the hormone ghrelin as playing a significant role in this interaction. In addition to playing a crucial function in the homeostatic control of energy metabolism, ghrelin and its receptor also influence the regulation of food-related behaviors, such as the hedonic rewarding and motivational pathways. A growing body of evidence points to the ghrelin system’s involvement in stress-related increases in appetite, emphasizing the significance of examining any possible connections between anxiety-related mechanisms and obesity.

### 3.1. Ghrelin and Its Impact on the Brain

Although it is not produced in the brain, ghrelin can cross the blood–brain barrier, bind to GHSR-1a there, and affect the central nervous system [73] (Figure 3). Ghrelin mainly activates the hypothalamic NPY/AgRP orexigenic pathway in the arcuate nucleus (ARC), where ghrelin receptors are highly concentrated, as well as the ventral tegmental area (VTA), nucleus accumbens, amygdala, and hippocampus [74]. Ghrelin appears to impact numerous other locations, including the amygdala, insula, orbitofrontal cortex, and striatum, all of which are involved in reward processing and appetitive [75].

Among the components of the ghrelinergic system are acyl-ghrelin, des-acyl ghrelin, a variety of alternative peptides (e.g., obestatin), GHSR-1a, and GOAT. The ghrelinergic system’s components have been connected to a broad spectrum of pathological processes and higher brain functions, including memory [76], reward [77], mood [78], and sleep [23]. Some of these functions are impaired in neurodegenerative illnesses, such as Parkinson’s disease (PD) [79], Alzheimer’s disease (AD) [80], and Huntington’s disease (HD) [81], all of which have been associated with ghrelin. Disturbances in the ghrelinergic system have also been linked to the development of stress-related mood disorders, such as depression and anxiety [82]. The precise mechanisms of how the ghrelinergic system impacts the brain are still under investigation and are likely multifactorial [83].

### 3.2. Ghrelin Gene Polymorphism in Psychiatric Disorders and Obesity

According to Nakashima et al., the Ghrl gene polymorphism is strongly associated with depression. There are notable differences in the frequency of the Leu72Met ghrelin gene variants between patients with major depressive disorder (MDD) and healthy controls [84]. However, there are no apparent differences between individuals with panic disorder and healthy controls [84]. A higher incidence of panic disorder may be associated with the Gln90Le polymorphism in the pre-proghrelin gene [85].

The G/G genotype of the A-604G SNP (single-nucleotide polymorphism) of the GHRL gene was found to be associated with altered serum ghrelin levels and obesity [86].

### 3.3. Ghrelin Levels in Psychiatric Diseases

It is unclear whether serum ghrelin levels are somehow altered in patients with depression. In the Ozsoy et al. study, depressive patients had higher serum ghrelin levels, which subsequently decreased to normal when their depressive symptoms subsided [87]. This finding is confirmed by other studies [88,89,90]. Kluge et al., however, found no significant differences in ghrelin blood concentration between patients with MDD and control volunteers [91]. It was also reported that ghrelin levels in depressed individuals might be reduced compared to those in controls [92].

It has been demonstrated that stress exposure changes ghrelin levels. For instance, in rodents, acute stresses, such as tail pinch stress and water avoidance stress, have been reported to increase gastric ghrelin gene expression and plasma ghrelin levels [93,94]. Interestingly, long-term stress increases the amount of circulating acyl-ghrelin, and this increase lasts long after the stressor stops [95]. Likewise, chronic exposure to stressors in human individuals also causes an increase in acyl-ghrelin, which lasts long after stressor cessation [96]. It is hypothesized that ghrelin secretion may be a counter-regulatory response to stress and that higher amounts of ghrelin may be necessary to prevent excessive levels of anxiety.

Exogenous ghrelin injections have been proven to provide anxiolytic- and antidepressant-like effects [95,97].

The bilateral removal of the olfactory bulbs in adult rodents causes several behavioral changes that are associated with altered stress sensitivity, which is regarded as a validated model of severe depression. According to a study by Carlini et al., mice who had had their bilateral olfactory bulbs removed had longer periods of inactivity in the tail suspension test than control animals, which is a sign of depressive-like behavior. Ghrelin 0.3 nmol/l administration reversed that immobility and encouraged escape-related behavior, indicating an antidepressant effect of this peptide [97].

These findings are consistent with those made by Lutter et al., who examined whether ghrelin signaling regulates depressive symptoms in a mouse model of chronic stress. GHSR-null mice demonstrated increased social isolation following chronic social defeat stress, showing that ghrelin signaling may be implicated in stress responses. Furthermore, a peripheral injection of ghrelin to mice resulted in anxiolytic and antidepressant-like reactions measured in the elevated plus-maze and the forced swim tests [95]. Several other studies have, however, demonstrated anxiogenic and depressant-like effects of ghrelin injections in rodents, which appears to be at odds with previously mentioned studies [93,98]. According to Carlini et al., ghrelin not only causes anxiogenesis but also increases memory retention in rats, indicating that the peptide may have an impact on hippocampus-related processes [99]. Furthermore, Jackson et al. suggested that intracerebroventricular ghrelin administration increases depressive-like behavior in male juvenile rats [100].

There is also some evidence that ghrelin levels correlate with diverse neurotransmitters, neuromodulators, and their receptors, like monoamines; neuropeptides; and endocannabinoids [101,102,103]. According to Brunetti et al., ghrelin inhibits serotonin release [103].

Some researchers suggest that ghrelin also affects neuroplasticity, in which abnormalities are intimately associated with depression [104] and may be linked to variations in neurotrophic factor levels, particularly BDNF [105]. Multiple studies and meta-analyses have found that people with depression had reduced BDNF levels in their blood [106,107,108], along with lower ghrelin levels. Lower ghrelin levels are thought to promote a reduction in BDNF levels and cognitive impairment in depression via the inhibition of the cAMP-signaling pathway in depressed male mice. Furthermore, ghrelin therapy was shown to cause dendritic spine remodeling in hippocampal neurons and promote the expression of certain BDNF-mRNA species [109].

### 3.4. Ghrelin Levels in Obesity

The ghrelin level in the blood is also related to the body’s energy status. There is a negative correlation between circulating ghrelin levels and body mass index [110,111]. Ghrelin levels are decreased in obesity [112] but increased in lean individuals, including those who suffer from anorexia nervosa [113] or cachexia [114]. Since ghrelin was found to enhance food intake in healthy individuals, it is possible to interpret the reduced ghrelin activity in obesity as a counter-regulatory mechanism to prevent further increases in food intake and body weight. Ghrelin levels in the blood increase during weight reduction [115] and decrease during weight gain [116,117,118], implying that ghrelin fluctuations are one of the compensatory mechanisms that keep the body’s energy balance stable.

### 3.5. Possible Mechanisms Linking Ghrelin to Psychiatric Disorders

Hyperactivity of the hypothalamic–pituitary–adrenocortical (HPA) system and chronic inflammation are two of the major pathophysiological factors for the development of depression [119,120]. Increased levels of proinflammatory cytokines and stress hormones, like glucocorticoids, have been shown in both experimental and clinical investigations to significantly contribute to the behavioral changes linked to depression. Moreover, there is a significant positive association between blood cortisol levels, depression, and Hamilton Depression Rating Scale scores [121]. 

Ghrelin primarily stimulates the HPA axis at the hypothalamus level [95] by stimulating vasopressin [122] and indirectly activating corticotropin-releasing hormone (CRH) neurons [123]. Additionally, ghrelin increases both hormone production and gene expression in the hypothalamus. Ghrelin has been demonstrated to enhance both the gene expression of corticotropin-releasing hormone (CRH) [93] and CRH release [124]. Furthermore, ghrelin raises serum levels of both ACTH [125] and cortisol [126].

It is clear that depression is associated with dysfunctions in the hypothalamic–pituitary–adrenal (HPA) axis. According to numerous studies, melancholic and endogenous depression subtypes have a greater correlation with enhanced HPA-axis activity than the atypical subtype. Some researchers hypothesize that, like PTSD, the atypical depression subtype may be linked to an opposite type of HPA axis dysfunction—hypofunction [127]. Ghrelin’s modulatory effect on the HPA axis may suggest that a changing ghrelin level contributes to the mechanisms responsible for the development of depression.

According to studies, ghrelin has significant anti-inflammatory capabilities, among others, through modulating the release of both proinflammatory and anti-inflammatory cytokines from LPS-stimulated macrophages via different signaling cascades. Exogenous ghrelin decreases the production of proinflammatory cytokines IL-1β and TNF-α in LPS-stimulated murine macrophages, presumably by decreasing LPS-induced NFκB activation. Dixit et al. confirmed that ghrelin suppressed the release of proinflammatory cytokines (IL-1, IL-6, and TNF-) in human monocytes, as well as in T cells and peripheral blood mononuclear cells [128].

Furthermore, exogenous ghrelin significantly increased anti-inflammatory IL-10 release from LPS-stimulated murine macrophages via the activation of p38 MAPK, which is known to influence IL-10 release in macrophages independently of the NFκB pathway. Interestingly, when a particular GHS-R receptor antagonist was introduced to the culture media, the effects of ghrelin on both proinflammatory and anti-inflammatory cytokines were abolished [129].

These findings suggest that ghrelin can reduce inflammation and is significant in both metabolic and nonmetabolic inflammatory diseases.

Ghrelin has also been shown to affect central vagus nerve activity. Wu et al. demonstrated that exogenous ghrelin reduces inflammatory cytokine TNF-α and IL-6 release via the activation of the vagus nerve in a mouse model of endotoxemia, implying that ghrelin may link the central nervous and immune systems [130].

According to Bansal et al., the release of ghrelin plays a crucial role in the anti-inflammatory effects of vagal nerve stimulation (VNS) after traumatic brain injury (TBI). Intestinal barrier breakdown following TBI is characterized by increased intestinal permeability, leading to bacterial translocation and inflammation. TNF-α levels after TBI can be reduced with both electric vagus nerve stimulation and exogenous ghrelin. Although vagus nerve stimulation increases serum ghrelin, which is primarily derived from the stomach, blocking the ghrelin receptor limits the anti-inflammatory effects of vagus nerve stimulation [131].

Given these findings and the available research, gastrointestinal hormones such as ghrelin appear to affect mood; nevertheless, the precise interplay between the nervous system, ghrelin regulation, and mental disorders requires further investigation.

### 3.6. Possible Mechanisms Linking Ghrelin to Obesity and Metabolic Disorders

According to researchers, ghrelin may be involved in the metabolism of insulin and glucose. In healthy subjects, acyl-ghrelin reduced insulin levels [132] while increasing glucose levels [133], whereas des-acyl-ghrelin enhanced glucose metabolism and insulin sensitivity [134]. GHS-R has been found to be expressed in pancreatic cells, and acyl-ghrelin has been shown to inhibit insulin release via a Ca^2+^-mediated pathway [135].

According to Pacifico et al.’s findings, obese children with metabolic syndrome showed lower levels of des-acyl-ghrelin and a higher acyl-ghrelin/des-acylghrelin ratio than obese children without metabolic syndrome [136]. In support of this discovery, Rodriguez et al. reported that obese people with normoglycemia and type 2 diabetes mellitus had higher plasma levels of acyl-ghrelin and lower levels of des-acylghrelin than lean individuals [137]. These findings may suggest that excessive acyl-ghrelin levels promote insulin resistance and obesity. Interestingly, Ozcan et al. discovered that a DAG injection improved glycemic control in obese type 2 diabetes patients by suppressing AG levels [138]. This could imply that DAG is a promising option for the creation of drugs for the treatment of metabolic illnesses or other medical conditions characterized by an abnormal AG:DAG ratio, such as type 2 diabetes or Prader–Willi syndrome.

Several elements of metabolism and inflammation are regulated by ghrelin and other factors, resulting in improved or worsening insulin resistance and metabolic syndrome [139]. Obesity-mediated metabolic abnormalities raise the levels of numerous cytokines and chemokines, resulting in a proinflammatory state, which is a possible risk factor for the development of inflammation-induced insulin resistance. Monocyte chemotactic protein-1 (MCP-1) is one of these secreted factors that is stimulated by the NF-B pathway to recruit monocytes, resulting in an increase in the M1 ATM macrophage population profile [140]. The ratio of M1-like to M2-like macrophages is significantly enhanced, resulting in an abundant secretion of proinflammatory mediators, such as TNF, IL-6, and IL-1 [141]. This imbalance, characterized by a proinflammatory profile, suppresses the insulin-signaling system and may result in metabolic syndrome.

Obesity-related ghrelin regulation dysfunction will lead to systemic inflammation, which, in turn, will promote neuroinflammation. This will cause microglial proliferation and astrocyte degeneration [142,143], which will promote kynurenine pathway activity and decrease the availability of tryptophan for serotonin synthesis [144]. Additionally, inflammation reduces BDNF secretion and hippocampus neurogenesis, both of which are frequently associated with depression [145].

However, it has been demonstrated that calorie restriction increases plasma acyl-ghrelin, which then improves hippocampus neurogenesis and provides a significant survival benefit to an organism. An increase in the number of new hippocampus neurons improves the chances of successful re-feeding and survival by improving the ability to remember the specific locations of food and identifying safe settings, especially during times of limited supplies [146].

Diet-induced obesity causes hypothalamic inflammation, leading to central ghrelin and insulin resistance. Due to modifications in homeostatic feeding circuits and reward-processing pathways, ghrelin can no longer increase food intake [147]. Ghrelin resistance occurs through lowering the NPY/AgRP neuronal response to ghrelin, as well as the hypothalamic expression of Ghsr [148]. According to Naznin’s research, peripheral ghrelin resistance is also linked to inflammation in the nodose ganglia of mice given a high-fat diet; this impairs the vagal afferent pathway [149], which can result in sympathetic overactivity [150,151,152,153].

According to Lutter, ghrelin resistance associated with obesity will decrease its antidepressant and neuroprotective effects, causing symptoms of MDD [154]. Moreover, stress-related ghrelin resistance, independent of hunger responses, appears to contribute to amygdala hyperactivation and the overconsolidation of fear memories in animals [155].

## 4. Ghrelin, and the Microbiota and Its Link to Immunometabolic Depression

From the stomach to the colon, the number of bacteria colonizing the gastrointestinal tract grows significantly. The main phyla of bacteria colonizing the human gastrointestinal tract are Firmicutes (60 to 80%), followed by Bacteroidetes (20 to 40%), Proteobacteria, and Actinobacteria (5%) [156,157].

Although a healthy person’s microbiome tends to remain stable, lifestyle and dietary habits can undoubtedly impact gut microbial dynamics. Eubiosis, often known as “healthy microbiota”, is the equilibrium of the intestinal microbial ecosystem with the majority of potentially helpful bacteria species. Dysbiosis, which means disturbance of the intestinal microbiota, is a condition that interferes with intestinal homeostasis, causes inflammation, and negatively affects intestinal permeability [158]. Changes in the gut microbiota negatively affect the gut–brain axis on numerous levels; the HPA axis becomes overactive, the intestinal barrier is impaired, the immune system produces excessive amounts of proinflammatory cytokines, and neurotransmitter levels are disrupted [159].

The interaction between the microbiome, the gastrointestinal tract, and the central nervous system is called the microbiota–gut–brain axis. Short-chain fatty acids (SCFAs) and neuroactive signaling molecules, like gamma-aminobutyric acid (GABA), serotonin, and dopamine, are just a few examples of the bioactive chemicals and metabolites that the microbiome produces and that have an impact on the brain [160]. In addition, the gut microbiota significantly impacts the production and secretion of gut peptides, including ghrelin [82].

### 4.1. Ghrelin Levels, and the Composition of Gut Microbiota and Its Link to “Metabolic Depression”

According to research, circulating ghrelin levels change alongside the composition of the gut microbiota, indicating that the commensal bacteria in the gut may regulate the ghrelinergic system [161]. Gut microbiota alterations are also linked to depressive disorder and correlate with the severity of a patient’s symptoms [162].

Studies have shown a positive association between ghrelin levels and Clostridium and Ruminococcus [163,164,165], which are often decreased in patients with depression [1,2]. 

An increased Bacteroidetes/Firmicutes ratio and Faecalibacterium [166], Prevotellaceae [167], and ghrelin levels have all been associated negatively. In depressed patients, these inverse correlations are also seen. Faecalibacterium [3] and Prevotellaceae [4] levels are reduced in patients with depression. Changes in the Bacteroidetes/Firmicutes ratio may also promote depression [5].

However, some associations are uncertain and vary across studies. For example, Bacteroides [168,169] and Bifidobacterium [169,170] are proven to have both positive and negative relationships with ghrelin. 

Changes in gut microbiota compositions and a decrease in ghrelin levels were also observed following H. pylori eradication treatment [171]. It is interesting to note that antibiotic eradication therapy for H. pylori infection is linked to a significant short-term (less than 30 days) rise in the prevalence of depressive disorder [6].

Both fasting and postprandial ghrelin levels have been linked to particular bacterial alterations. It has been reported that food restriction promotes the growth of mucin-degrading bacteria (for example, Prevotella); this may result in an increase in mucin degradation and a lack of mucin on the gut epithelial layer, which would negatively affect intestinal permeability and ghrelin levels [172].

### 4.2. Microbiota Metabolites and Ghrelin Connection

The host offers a stable environment for microorganisms, while microbes provide the host with a variety of services, such as the digestion of complex dietary macronutrients, nutrient and vitamin synthesis, pathogen protection, and immune system maintenance. SCFAs, particularly acetate, propionate, and butyrate, are the main products of microbial fermentative activity in the gut [173]. 

According to research by Perry et al., changed gut microbiota that produces more acetate activates the parasympathetic nervous system, which, in turn, stimulates hyperphagia, increased ghrelin secretion, increased glucose-stimulated insulin secretion, and obesity and its associated implications [174]. In terms of evolution, this was an adaptation for foraging animals; but, nowadays, when people are exposed to calorically dense, abundant food regularly, this promotes obesity and its associated sequelae.

Torres-Fuentes et al. demonstrated that bacteria-derived SCFAs (acetate, propionate, and butyrate) and lactate, as well as bacterial supernatants from the Bifidobacterium and Lactobacillus genera, may affect GHSR-1a signaling. Butyrate, propionate, acetate, and lactate were all found to be effective at inhibiting ghrelin-mediated calcium mobilization in GHSR-1a-expressing cells. This indicates the potential of commensal bacteria metabolites as major components of host interactions via the modulation of host G Protein Coupled Receptor (GPCR) signaling and alludes to a significant unique functionality of microbiota-derived SCFAs in gut–brain axis signaling [175].

These findings emphasize the gastrointestinal microbiota’s significant therapeutic potential as a rich supply of metabolites targeting specific GPCRs within the gut–brain axis, impacting both the CNS and peripheral nervous system. The gut microbiota and its metabolites show considerable promise in the development of novel treatment techniques to treat a variety of metabolic and mental illnesses involving GHSR-1a signaling. It is important to note that bacterial supernatants from the same genera (i.e., Bifidobacterium species) might have radically distinct effects, showing that probiotic effects are species- and strain-dependent [175].

## 5. Ghrelin-Related Factors in Diagnosis, Prevention, and Treatment of Immunometabolic Depression

The potential role of ghrelin in the biological link between obesity and stress may help in the development of a more effective preventive strategy.

Ghrelin may act as a biomarker of stress, according to research by Bouillon-Minois et al., who demonstrated that the hormone increases briefly after an acute stress intervention. Furthermore, overweight and obese people showed a prolonged ghrelin response compared to normal-weight individuals, suggesting a relationship between obesity and stress [176]. The association between the increased and prolonged ghrelin response to stress in overweight and obese people requires further study. This association between obesity and stress proposes that stress management programs may be implemented in overweight avoidance and obesity for long-term weight loss. The findings of that study might support the use of ghrelin as a screening tool to identify individuals who are at risk of obesity and stress- related disorders.

Because of the multiple favorable effects on system metabolism, pharmacological manipulation of the endogenous ghrelin system is widely regarded as a viable method for treating a range of metabolic diseases, including obesity and metabolic disorder. Because ghrelin administration promotes food intake and body weight growth, restricting ghrelin action is suggested to prevent body weight gain and adiposity. Indeed, the neutralization of circulating ghrelin by antibodies and RNA Spiegelmers, the antagonism of the ghrelin receptor by antagonists and inverse agonists, the inhibition of ghrelin O-acyltransferase (GOAT), and potential pharmacological approaches to decrease ghrelin synthesis and secretion are all therapeutic strategies for metabolic disorders targeting the ghrelin pathway [177].

D-Lys-3-GHRP-6, a peripherally administered GHS-R antagonist, reduced food intake in lean mice, diet-induced obese mice, and obese mice. Furthermore, the repeated administration of D-Lys-3-GHRP-6 reduced body weight gain and enhanced glycemic control in obese mice. In contrast, the repeated treatment of ghrelin, an endogenous ligand for GHS-R, increased obesity and impaired glycemic control in a high-fat diet [178]. 

Inverse agonists of the ghrelin receptor may also be useful in the treatment of obesity-related metabolic disorders. GHSR-IA1 inverse agonist administration in Zucker diabetic fatty rats enhanced the metabolic rate, decreased food intake, and improved oral glucose tolerance. Furthermore, an injection of a GHSR-IA2 inverse agonist reduced food intake and body weight in diet-induced obese mice, as well as blood lipids [179]. F-05190457 is the first inverse agonist of the ghrelin receptor tested in humans, and it has been proven to suppress growth hormone release and gastric emptying and to reduce postprandial glucose levels; however, its effects on body weight have not been examined [180]. Ghrelin *signaling* neutralization is recognized as one of the most promising therapeutic therapies for obesity. In 2006, Shearman et al. administered in vivo a novel type of ghrelin-blocking agent, the RNA SPM NOX-B11-2, which inhibited ghrelin function in vitro. In diet-induced obese mice, researchers found that selective ghrelin inhibition successfully promoted fat and weight loss and reduced food consumption [181]. The concept of countering ghrelin activity has been used to develop vaccinations that promote the creation of anti-ghrelin antibodies. This was thought to reduce food efficiency and fat growth, but the evidence is inconsistent [182] Aside from hyperphagia and obesity, ghrelin is also heavily engaged in the control of glucose metabolism. As a result, there has been some interest in the potential of ghrelin antagonism to treat diabetes and hyperglycemia. Ghrelin administration decreases insulin sensitivity; limits insulin secretion; elevates blood cortisol; and stimulates the release of glucagon, somatostatin, and growth hormone, among other known regulatory actions that may help explain its ability to raise glucose levels. In contrast, diet-induced obese mice with a persistent pharmacological blockage of GHSRs and the genetic deletion of other ghrelin system components that typically activate GHSR signaling show improved glucose tolerance and/or insulin sensitivity [183,184]. Studies have indicated that GHSR antagonists enhance insulin secretion and glucose tolerance in obese individuals [185], and more recent research has revealed that ghrelin inhibition using the GHSR antagonist GHRP-6 can reverse the symptoms of diabetes [186]. 

The significance and therapeutic potential of the ghrelin system in mental disorders are also gaining attention. Although the ghrelin system’s role in stress-related psychiatric diseases is still not fully understood, it offers a novel perspective in the pursuit for more effective diagnostic and treatment approaches [83]. Future ghrelin-based treatments could make use of GHSR agonists, antagonists, inverse agonists, GOAT inhibitors, and other compounds. As far as we know, there have not been any studies published that specifically investigate the impact of ghrelin-targeted therapy in stress-related psychiatric illnesses. More study is required since ghrelin has the potential to be a therapeutic target as well as a biomarker of stress-related illnesses.

Additionally, as changes in food and the composition of the gut microbiome have been associated with variations in plasma ghrelin levels, indirect targeting of the ghrelin system by means of other approaches, such as the gut microbiome, may be relevant in stress-related psychiatric disorders [187]. This emphasizes the potential of using the gut microbiome to affect ghrelin signaling and thus modify the gut–brain axis under conditions of stress.

## 6. Conclusions

Numerous studies indicate that ghrelin plays a significant role in the neural circuitry underlying the complex correlation between stress, mood, food intake, and obesity (Figure 4). Ghrelin secretion is altered in a variety of psychiatric and metabolic disorders, although the relationship between these changes is unclear and most likely multifactorial. The ghrelinergic system is thought to be a promising target for developing efficient pharmacotherapies since it not only influences obesity but also has an impact on psychological well-being under conditions of stress, anxiety, and depression.

## Figures and Tables

**Figure 1 nutrients-15-03960-f001:**
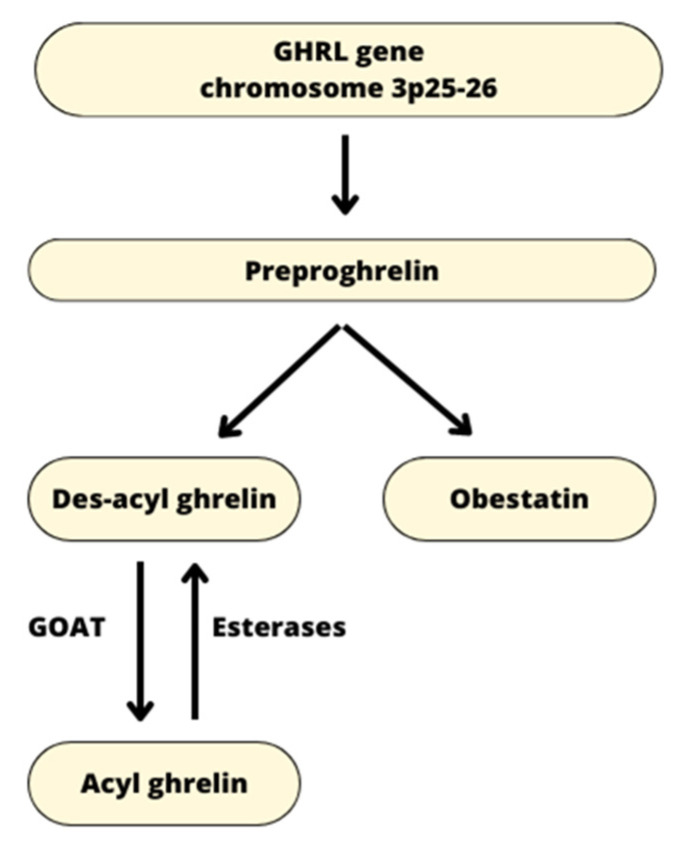
A schematic graphic depicting the successive steps in the production of ghrelin gene products and their connection with GOAT. The ghrelin gene in humans is located on chromosome 3p25-26. The GHRL gene encodes the preproghrelin peptide, which is later transformed into des-acyl ghrelin (DAG) and obestatin. DAG is then modified enzymatically in the endoplasmic reticulum by GOAT to produce acyl-ghrelin. GOAT—ghrelin-o-acyltransferase.

**Figure 2 nutrients-15-03960-f002:**
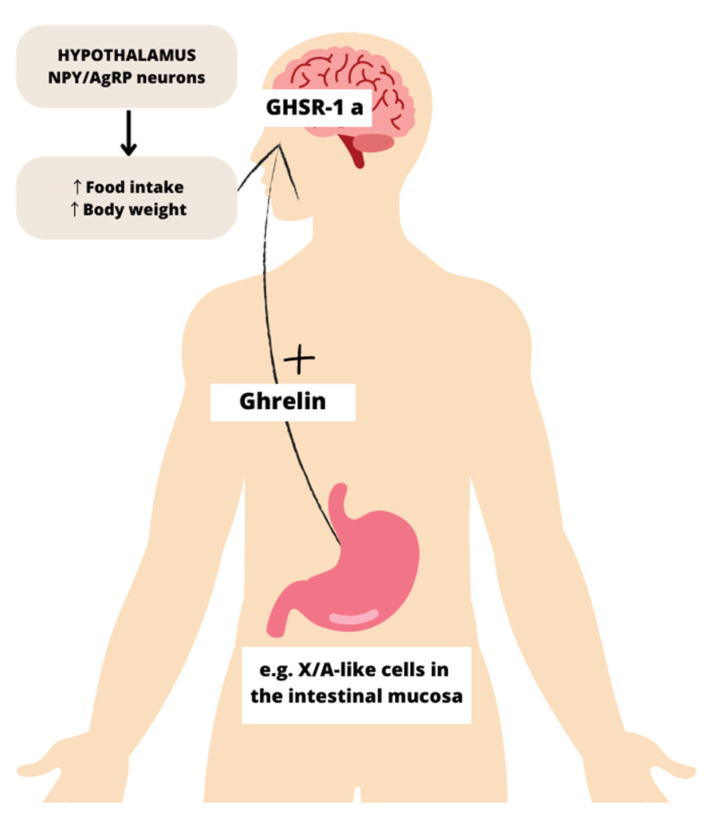
Schematic diagram of the ghrelin signal pathway. Ghrelin, which is produced by X/A-like cells in the intestinal mucosa, binds in the hypothalamic arcuate nucleus to the growth hormone secretagogue receptor (GHSR) in neuropeptide Y (NPY) and agouti-related peptide (AgRP) neurons, subsequently increasing hunger and food intake. GHSR-1a—growth hormone secretagogue receptor; NPY/AgRP neurons—neuropeptide Y/agouti-related peptide neurons.

**Figure 3 nutrients-15-03960-f003:**
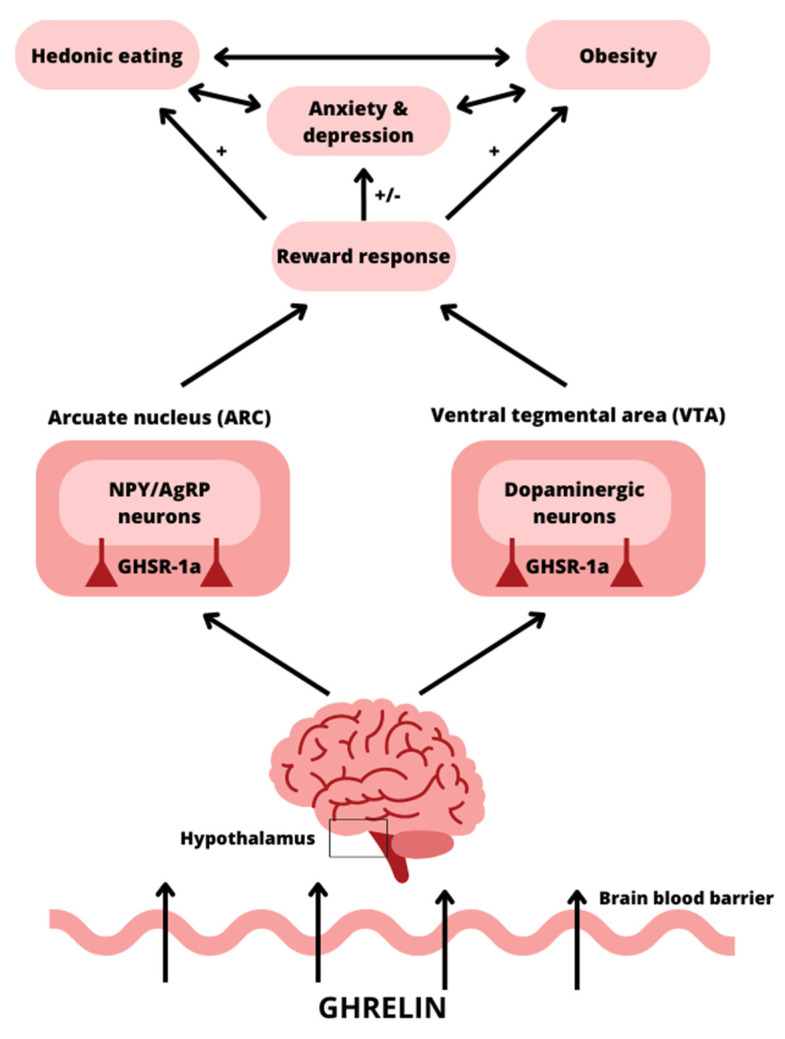
A schematic diagram demonstrating ghrelin’s effect on two major brain regions, namely, the hypothalamic arcuate nucleus (ARC) and the ventral tegmental area (VTA), and its relationship with reward response, obesity, hedonic eating, anxiety, and depression. Plasma ghrelin levels rise in response to psychological stress, activating the hedonic signaling pathway and promoting the consumption of food. This stress-related hyperphagia and food reward behavior lead to an increase in body weight, which has been associated with major depressive disorder and anxiety. NPY/AgRP neurons—neuropeptide Y/agouti-related peptide neurons. GHSR-1a—growth hormone secretagogue receptor; +—stimulating; −—inhibiting.

**Figure 4 nutrients-15-03960-f004:**
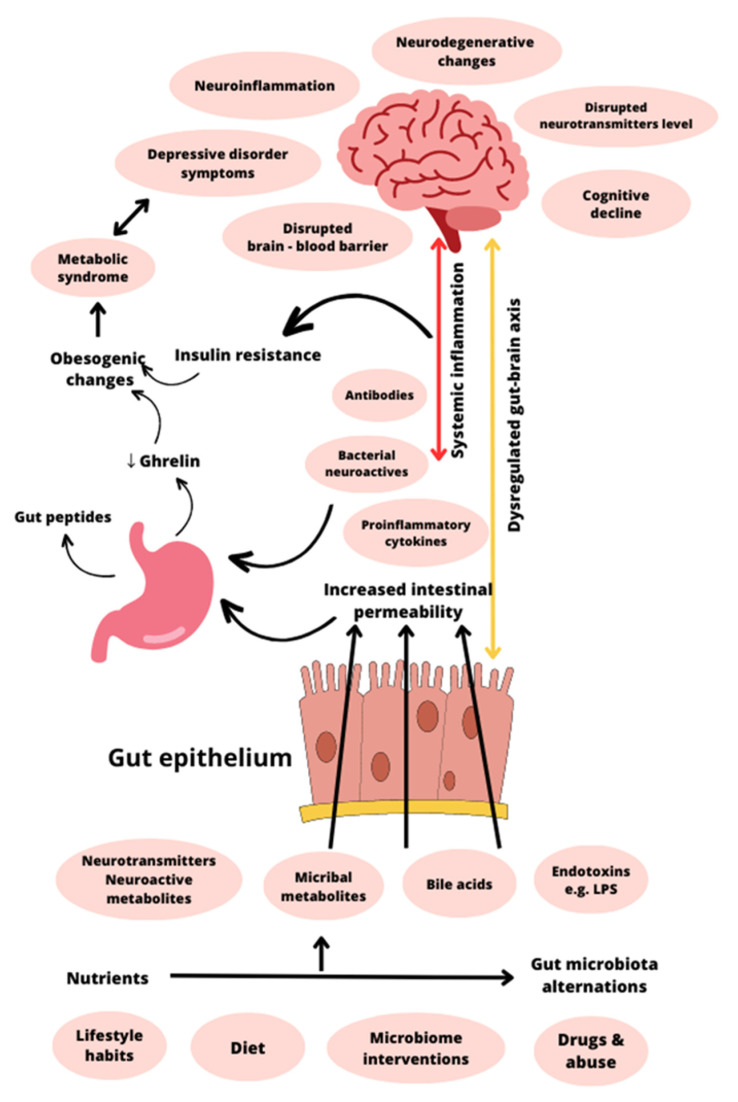
A schematic model summarizing the proposed processes linking ghrelin signaling, inflammation, and microbiota in pathological states like depression and metabolic syndrome. Lifestyle habits, diet, microbiome intervention, drugs and abuse, and nutrition can lead to specific gut microbiota alternations. The microbiota and its metabolites may affect the gut–brain axis via a variety of mechanisms, including enteroendocrine cells, the vagus nerve, and the enteric nervous system. Increased gut permeability due to altered gut microbial composition induces a proinflammatory state, which could lead to neuroinflammation with all its consequences, for example, depression and anxiety. Proinflammatory conditions cause insulin resistance and decreased ghrelin levels, which may result in obesity. Obesogenic changes also contribute to depression and anxiety. LPS—lipopolysaccharide; ↓-decreased level of.

**Table 1 nutrients-15-03960-t001:** The table shows the basic physiological functions of ghrelin in relation to various organs.

Organ	Ghrelin’s Physiological Effects	Sources
Hypothalamus	↑ Appetite	[18]
	↑ Food intake	[19]
	Reward behavior	[20]
	Olfaction and sniffing	[21]
	Learning and memory	[22]
	Depression	[23]
	Sleep/wake rhythm	[23]
Pituitary	↑ Growth hormone	[1,24]
	↑ ACTH	[25]
Sympathetic nervous system	Modulation of the sympathetic nervous system	[26]
Brown adipose tissues	↓ Thermogenesis	[27]
Pancreas	↓ Insulin secretion	[28]
	Modulation of insulin sensitivity	[29]
	Glucose metabolism	[30]
Heart	↑ Cardiac output	[31,32]
	↑ Vasodilatation	[33]
Liver	↑ IGF-1	[34]
Stomach	↑ Gastric emptying and ↑ acid secretion	[35,36]
Intestine	↑ Intestinal motility	[37]
Adipose tissue	↑ Lipogenesis	[38]
	Regulation of inflammation	[39]

ACTH—adrenocorticotropic hormone; IGF-1—insulin-like growth factor 1; ↑—increasing level of; ↓—decreasing level of.

## Data Availability

Data sharing not applicable.

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
