# Peer review of "Ghrelin as a Biomarker of “Immunometabolic Depression” and Its Connection with Dysbiosis"

_nutrients, 2023, doi:10.3390/nu15183960_

Round 1

Reviewer 1 Report

This review manuscript reported the growing research revealing the relationship between ghrelin, diet, microbiome, and disorders like depression and metabolic syndrome. It also highlighted the increasing research established the close relationship between ghrelin, nutrition, microbiota, and disorders such as depression and metabolic syndrome, and will evaluate the ghrelinergic system as a potential target for the development of effective pharmacotherapies. 

1.          There is many information presented in the manuscript but it is too messy, and needs to be reorganized to make it clear for readers to understand. 2.          The abbreviations must define at the first time showed in the manuscript. 3.        The meaning of ↑ and ↓ in Table 1 must explain in the table. 4.        Figures are helpful for readers to understand the concept of this manuscript. The Figure Legends have to clear present the content of the Figure. 5.          Many references are cited in this manuscript, however, more paper published in recent two years must cited and describe in this review. 6.        All the form of reference cited in the text and “Reference” section in this manuscript have to follow the format of Nutrients.

7.        The are some type and grammar errors in the manuscript. Please recheck the manuscript carefully. 1.        The are some type and grammar errors in the manuscript. Please recheck the manuscript carefully.

Reviewer 2 Report

This study is expected to provide valuable information on the relevance of ghrelin to immune dysfunction and intestinal bacterial imbalance.

Paragraphs are excessively divided in the paragraph composition part. Please supplement this part.

The key part of this study is that ghrerin is related to immune function decline. This part needs to be described in more detail in this study. In addition, there is a need to organize and present the contents of this part in more detail.

The conclusion of this study should also include key information on the relationship between the effect of ghrerin on the decline in immune function and the imbalance of intestinal bacteria.

Moderate editing English language is required.
